# Reliability of Different Anterior Cranial Base Reference Areas for Voxel-Based Superimposition

**DOI:** 10.3390/jcm10225429

**Published:** 2021-11-20

**Authors:** Mohammed Ghamri, Georgios Kanavakis, Nikolaos Gkantidis

**Affiliations:** 1Department of Orthodontics and Dentofacial Orthopedics, University of Bern, 3010 Bern, Switzerland; mghamri@moh.gov.sa; 2Directorate of Health Affairs-Jeddah, Ministry of Health, Riyadh 11176, Saudi Arabia; 3Department of Pediatric Oral Health and Orthodontics, UZB-University Center for Dental Medicine, University of Basel, 4058 Basel, Switzerland; georgios.kanavakis@unibas.ch; 4Department of Orthodontics, Tufts University School of Dental Medicine, Boston, MA 02111, USA

**Keywords:** imaging, three-dimensional, cone-beam computed tomography, voxel-based superimposition, anterior cranial base

## Abstract

The study aimed to evaluate the reliability and reproducibility and compare the outcomes of two 3D voxel-based superimposition techniques for craniofacial CBCT images, using anterior cranial base areas of different extent as references. Fifteen preexisting pairs of serial CBCTs (initial age: 11.7 ± 0.6 years; interval: 1.7 ± 0.4 years) were superimposed on total anterior cranial base (TACB) or middle anterior cranial base (MACB) structures through the Dolphin 3D software. The overlap of the reference structures was assessed visually to indicate reliability. All superimpositions were repeated by the same investigator. Outcomes were compared to assess the agreement between the two methods. Reliability was perfect for the TACB and moderate for the MACB method (*p* = 0.044). Both areas showed good overall reproducibility, though in individual cases there were notable differences for MACB superimpositions, ranging from −1.84 to 1.64 mm (TACB range: −0.48 to 0.31 mm). The overall agreement in the detected T0/T1 changes was also good, though it was significantly reduced for individual measurements (median < 0.01 mm, IQR: 0.46 mm, range: −2.81 to 0.73 mm). In conclusion, the voxel-based superimposition on TACB was more reliable and showed higher reproducibility than the superimposition on MACB. Thus, the extended anterior cranial base area is recommended for the assessment of craniofacial changes.

## 1. Introduction

Craniofacial growth assessment and treatment outcome analysis is essential in several fields, such as orthodontics and maxillofacial or plastic surgery. The most commonly used method for this is the superimposition of serial 2D cephalometric images on stable areas of the anterior cranial base [1]. However, the radiographic source of information is in two dimensions, which has limitations, such as image distortion and lack of precision in landmark identification. These are mainly related to the compression of a 3D object into a 2D image [1,2] and impact the reliability of the obtained information.

Therefore, the popularity of cone-beam computed tomography (CBCT) in diagnostics has increased over the years. The realistic representation of anatomical structures makes CBCT a powerful tool to diagnose and assess craniofacial changes, primarily due to the relatively low radiation, the adequate image quality, and the three-dimensional nature of the acquired information [3,4].

Recently, there is an increased interest in the use of serial CBCT superimpositions to evaluate craniofacial changes [5]. The anterior cranial base structures remain stable after an early age [6,7]. Therefore, and due to their being centrally located in the cranium, superimposition on these structures is the standard reference, applicable in 2D as well as in 3D images [4,5,8]. For 3D superimposition on the anterior cranial base, different methods and techniques have been proposed, with voxel-based, landmark-based, and surface-based superimpositions being the most commonly used [4]. Landmark-based superimposition depends on landmark identification. A considerable number of landmarks set by trained operators is required to achieve adequate precision, which needs more working time and increases costs [8,9]. For surface-based superimposition, a preliminary step is required to extract the surface models from the radiographic volume through bone segmentation. The superimposition outcome can be influenced by this process because it relies on grayscale value thresholds used for the segmentation [10,11], and in CBCTs, these do not directly correspond to Hounsfield units as in the CT images. Even if an image is acquired from the same machine, with a congruent setting, reproducibility in grey-level intensities cannot be guaranteed [12]. The third method is voxel-based superimposition, which is based on the best-fit approximation of the voxel greyscale values of corresponding areas, selected directly on the original volumetric data. Thus, this method overcomes the limitations of the previous methods, though for an easier assessment of the 3D outcome, surface models still need to be extracted from voxel-based superimposed volumes. Nevertheless, the latter applies to the visualization of the outcomes and not to the superimposition process per se, which requires no segmentation.

The voxel-based craniofacial CBCT superimposition has been reported as a new method, where volume images and slices superimpositions are based on a best-fit concept, with mutual information as a matching criterion [13]. Cevidanes et al. have widely applied 3D voxel-based superimpositions for craniofacial changes assessment in different patient groups [14,15]. Bazina et al. tested the reliability of a similar approach in non-growing patients, using an easily operated software (Dolphin 3D software; Dolphin Imaging & Management Solutions, Chatsworth, Calif) and considering the anterior cranial base as a reference [16]. Weissheimer et al. [17] tested a 3D superimposition method on the anterior cranial base using OnDemand 3D software, while assessing the superimposition outcomes through ITK-SNAP software. Within their limitations, the above reports seemed to provide encouraging results [5]. In a recent previous study, the reliability of the anterior cranial base voxel-based superimposition was thoroughly tested in growing patients and exhibited satisfactory outcomes with respect to efficiency, cranial base matching, and reproducibility [18]. As previous reports [13,16,17], the latter study also used the entire anterior cranial base as a superimposition reference. Nevertheless, the midline anterior cranial base structures alone have also been used in the past and showed satisfactory results [19]. This area could be advantageous over the entire anterior cranial base, since the midline structures attain adult form (shape and size) earlier during development and show less interindividual variation, compared to the lateral structures [20,21,22]. A comparative assessment of voxel-based superimpositions on these two cranial base areas has yet not been performed. 

Therefore, in the present study we evaluated the agreement and reproducibility of a 3D voxel-based superimposition on two different anterior cranial base reference areas. The first was the total anterior cranial base (TACB), and the second was the middle anterior cranial base (MACB).

## 2. Materials and Methods

### 2.1. Ethical Approval and Study Design

This project was registered and approved by the Swiss Ethics Committees (Protocol No. 2018-01670). The study is a prospective methodological study, using pre-existing patient data, was conducted according to the guidelines of the Declaration of Helsinki, and participants signed an informed consent prior to the use of their data in the study.

### 2.2. Sample

Serial craniofacial CBCT images of 15 orthodontic patients (8 males, 7 females) were used in this study. The mean age of the participants at T0 (first acquired CBCT) was 11.75 ± 0.59 years, with a time lap between T0 and T1 (second acquired CBCT) of 1.69 ± 0.37 years. All subjects originated from a previously selected sample studied for the purpose of analogous projects [18,23], and the sample size is considered adequate [8,18,23].

Patients with systemic diseases, congenital malformation, or syndromes that could affect the facial morphology, as well as individuals with extreme facial asymmetries were excluded. Images with metallic restorative materials causing considerable artifacts and low-quality scans were also excluded. Two researchers (N.G, M.G) visually inspected all criteria independently to assess eligibility.

### 2.3. Generation of the CBCTs

As described previously [18,23], all tested CBCTs had been acquired in an orthodontic clinic between 2008 and 2018. The CBCTs images were acquired in cases where 3D information was needed to facilitate proper diagnosis and guide clinical decisions, such as in cases of impacted teeth. All CBCT images were acquired with the same X-ray machine (KaVo 3D eXam, Hatfield, PA, USA) under the following settings: 170 mm height × 232 mm diameter field of view, 0.4 mm^3^ voxel size, 5 mA tube current, 120 kV tube voltage, 8.9 s scan time, 3.7 s exposure time, which allowed for lower radiation doses [24]. The volumes were saved and exported in a DICOM format.

### 2.4. Superimposition Process and Reliability Assessment

Dolphin 3D software© (Version 2.1.6079.17633, Dolphin Imaging and Management Solutions Chatsworth, CA, USA) was used for voxel-based superimposition of the serial CBCTs. The pairs of DICOM datasets, obtained at two different time points (T0 and T1), were imported into the software. 

Two areas with different lateral extension were defined on the anterior cranial base as superimposition references. The first area represented the total anterior cranial base (TACB), while the second represented the middle anterior cranial base (MACB). Both areas included the midline anterior cranial base structures, which comprise a standard reference to assess craniofacial tissue changes [6,8,16].

The boundaries of the TACB area were defined by the posterior wall of sinus frontalis (anteriorly) to the middle of sella turcica (posteriorly). The height of the frame was determined at 3–4 cm, and the lower limit was set 2–4 mm inferiorly to the lowest point of sella turcica. The lateral limit of the selected frame was the lateral cranial wall. The boundaries of the MACB selection frame were defined previously [19] and included the posterior wall of the sinus frontalis (anteriorly) and the middle of the sella turcica (posteriorly) and were set inferiorly 2–4 mm below the lowest point of sella turcica, as for TACB, but the superior limit was located 1 cm above the anterior clinoid process. Additionally, the lateral extension of the frame was delimited by the width of the anterior wall of sella turcica (Appendix A).

CBCT T0 was always considered the base volume, where the reference structures were selected in a multiplanar view. Initially, CBCT T1 was manually adjusted and approximated to the base volume (CBCT T0) and then it was superimposed on it. The automated registration was repeated a few times (usually 2–3) till there was no visually identifiable change in the position of the superimposed volumes. The superimposition outcome was evaluated visually, checking the overlap of the anterior cranial base reference structures in 2D DICOM images, in the three planes of space (axial, sagittal, coronal). In case of inadequate overlap, the operator was satisfied with the maximum possible overlap achieved at the superimposed anterior cranial base structures. The superimposed CBCT T1 was then saved in its reoriented position, and the final visual assessment of the overlap of the superimposed anterior cranial base structures was recorded as a reliability measure.

### 2.5. Measurement Process

Following TACB and MACB superimposition of T1 to the corresponding T0 volumes, skeletal surface models were extracted from the T0 and T1 volumes through an automated bone segmentation function available in Dolphin Software. These were saved as STL files and imported in Viewbox 4 Software (version 4.1.0.1, BETA 64; dHAL software, Kifisia, Greece) to assess their differences.

To quantify differences in the detected T0–T1 changes between and within methods and compare errors in different areas of the facial skeleton, we used seven measurement areas, evenly distributed on the entire face, as previously published [18]. These areas were the N-point, A-point, Pogonion, Zygomatic arch right and left, and Gonial angle right and left. The seven areas were consistently selected on the extracted T0 surface model of every subject only once. Then, the T0 surface models with the seven areas were duplicated as needed to be used for all outcomes measured in the study. The size of each area was 100 triangles.

### 2.6. Intra-Operator Reproducibility of Superimposition Methods 

To test intra-operator reproducibility of the TACB and MACB superimpositions, one trained operator (M.G.) repeated the whole T0–T1 superimposition process for both methods. Following each superimposition, the STL files of all hard tissue surface models were imported into Viewbox 4 software, and the mean absolute distances (MAD) between corresponding T0–T1 surface models at the seven areas of interest were measured and compared.

Further assessment of the intra-operator agreement on the superimposition outcome was done through color-coded maps that showed the distances between the corresponding T1 surface models, following each repeated superimposition on the stable T0 model. 

### 2.7. Agreement between TACB and MACB

Following TACB and MACB superimpositions of T1 to the corresponding T0 volumes, the detected T0–T1 changes with each method at the seven measurement areas were recorded and compared.

Agreement between TACB and MACB methods was also assessed with color-coded distance maps between corresponding T1 models, after superimposition. Zero distance between these models would indicate perfect agreement.

### 2.8. Statistical Analysis

Statistical analysis was carried out with SPSS Software (IBM SPSS Statistics for Windows, Version 28.0. IBM Corp, Armonk, NY, USA). 

Data were tested with the Shapiro–Wilk test and were not found normally distributed in all cases. Thus, non-parametric statistical tests were used. Differences in reliability between the two methods were tested using McNemar’s test with the continuity correction. Intra-operator reproducibility of MACB and TACB superimposition outcomes is shown with box plots, where any deviation from 0 indicates superimposition error. Following MACB and TACB superimposition, differences in the amount of error (intra-operator reproducibility) among the different measured areas were tested in a paired manner through Friedman’s test. In case of significant results, pairwise comparisons were performed through Wilcoxon’s signed-rank test.

Differences in the detected T0–T1 changes between the MACB and the TACB superimpositions were visualized and tested in a similar manner.

In all cases, a two-sided significance test was carried out at an alpha level of 0.05. In case of multiple comparisons, a Bonferroni correction was applied to the level of significance to avoid false positive results. 

The Bland–Altman method (difference plot) [25] was also used to evaluate intra-operator reproducibility in the detected T0–T1 morphological changes through MACB, as well as the agreement between MACB and TACB superimpositions. Bland–Altman plots regarding the reproducibility of the TACB method have been published previously [18]. A one-sample *t*-test was used to assess if there was a systematic error between the compared measurements.

## 3. Results

### 3.1. Reliability of Superimposition Methods

When the TACB superimposition area was applied, adequate overlap of the anterior cranial base structures was visually observed in all 15 cases, in all three planes that were visualized in the 2D views of the software. On the contrary, when the MACB superimposition area was used, there were five cases that showed reduced overlap in the coronal and in the axial 2D view (Appendix A). In the sagittal 2D view, all cases showed adequate overlap. The difference in the reliability of the two methods was statistically significant (McNemar’s test: *p* = 0.044). Similar findings were evident when all assessments were repeated by the same operator.

### 3.2. Intra-Operator Reproducibility of Superimposition Methods 

For both TACB and MACB methods, the one-sample *t*-test demonstrated no systematic differences in the T0–T1 changes detected following repeated superimpositions by the same operator (*p* > 0.01). In the MACB superimposition, the median error was −0.02 mm (IQR: 0.23 mm), whereas in the TACB, the median error was <0.01 mm (IQR: 0.07 mm), which were both considered clinically irrelevant. However, when considering individual measurements, in the MACB superimposition, the deviations were mostly within 0.5 mm, but there were cases, especially considering the Gonial L area, where the differences were greater (Figure 1, Figure 2 and Figure 3). On the contrary, in the TACB superimposition, all individual differences remained within 0.5 mm (Figure 1 and Figure 2). There was no evidence for the MACB method indicating that the differences increased depending on the size of the detected T0–T1 changes. The differences between repeated MACB and TACB superimpositions tended to increase as the distance of the measurement area from the cranial base increased, except from the A-point area (Figure 1). No significant difference was detected between the magnitude of error at the various measurement areas (MACB, Friedman test: *p* = 0.130; Wilcoxon signed rank test: *p* > 0.008; Bonferroni corrected *p*-value = 0.002; TACB, Friedman test: *p* = 0.501).

Figure 2 shows the MACB and TACB intra-operator reproducibility through the color-coded distance maps of three cases for each method, representative of the minimum, the average, and the maximum error. It is evident that after MACB superimposition, the average error exceeded 1 mm in certain areas.

### 3.3. Agreement between TACB and MACB

Overall, the median agreement between the two methods was perfect (median < 0.01 mm, IQR: 0.46 mm, range: −2.81 to 0.73 mm; Friedman test: *p* = 0.084). However, for individual measurements, there were higher differences, even around 2.5 mm (Figure 4 and Figure 5). No significant difference was detected between the measurement areas shown in Figure 4 (Friedman test: *p* = 0.084; Wilcoxon signed rank test: *p* > 0.012; Bonferroni-corrected *p*-value = 0.002). There was no evidence from the Bland–Altman plots that the differences between methods were increasing by the amount of the measured T0–T1 change (Figure 5). The one-sample *t*-test demonstrated no systematic differences in the T0–T1 changes detected by the two methods (*p* > 0.01).

The color-coded distance maps provided in Figure 6 show individual cases representative of the minimum, the average, and the maximum differences of the relocated T1 surface models, superimposed on stable T0 models with both methods. It is evident that in all three cases, there are certain areas with differences around 2 mm, which can be considered clinically significant.

## 4. Discussion

Although the Dolphin 3D voxel-based superimposition method has been previously tested by different groups and the reports were encouraging [16,18,19], the effect of the extent of the anterior cranial base reference area on the superimposition outcomes was tested for first time here. Regarding average measurements, both TACB and MACB methods provided similar outcomes in the present growing patient sample. Furthermore, in the majority of the cases, the reproducibility of the outcomes of both areas was within 0.5 mm, except for a few cases superimposed on the MACB area. The findings confirmed that the method is user-friendly, fast, reproducible, and potentially reliable, which is in accordance with previous reports [16,18,19]. However, when individual measurements were considered, relatively large differences between the two methods became evident. In agreement with previous findings [18], the TACB method was proved reliable through the visual assessment of the overlap of stable ACB structures, as well as highly reproducible. On the contrary, the MACB method showed reduced reliability in about one-third of the cases. Furthermore, its reproducibility was reduced compared to that of the TACB method, and the two methods showed considerable differences, in regard to individual outcomes. Based on the above arguments, the MACB method cannot be recommended at present.

The MACB method has been previously introduced as sufficiently reproducible [19], but that study, among other limitations, did not assess individual differences between repeated measurements [5]. Our study performed a thorough assessment of the reliability and the reproducibility of this method, as well as of its agreement with the TACB method, and does not align with the aforementioned conclusions. Thus, despite the fact that the MACB method includes only the midline anterior cranial base structures that remain stable after approximately 7 years of age, it does not produce consistently reliable results. This is probably attributed to the relatively small size of the reference area that is used to register the corresponding volumes and not to the rationale of the area selection, which has a solid biological basis [6,7,20,21,22]. 3D surface superimpositions on small reference areas are less robust to artifacts and more prone to error [26]. A similar effect might be evident for the voxel-based registration of CBCT images, which is also based on best-fit registration algorithms. On the other hand, in accordance with previous reports [16,18], the present study confirmed that the TACB method offers sufficient reliability and higher reproducibility than the MACB method, despite extending to lateral anterior cranial base structures that attain anatomical form stability later in development, after approximately 11 years of age [21].

The level of threshold used to perform bone segmentation from 3D radiographic volumes has a significant impact on the extracted surface models, which could affect the validity of 3D superimposition outcomes [11]. Regarding outcome assessment, the segmentation error and the differences between manual and automated segmentations of a 3D dataset were tested in a previous study [18], and the amount of error was considered minimal in the majority of cases. In the present study, we used the automated function of Dolphin 3D software to extract the skeletal surface models from a specific volume, using always the same threshold [18], which eliminated the impact of the segmentation error on the tested outcomes. 

Although the study showed good overall reproducibility of the voxel-based 3D superimposition for both anterior cranial base reference areas, a critical observation of a few individual cases showed that when the MACB method was used, there were certain areas where the error exceeded 1 mm. This can be considered as clinically significant, especially if the segmentation error is added to the superimposition error. In voxel-based superimpositions, there is no segmentation error in the superimposition process itself, but this error is introduced in a second phase, since a proper 3D assessment of the outcomes requires the extraction of 3D surface models from the original radiographic volumes [11,18,23]. The outliers detected in the MACB method might be associated with the small field of the selected reference structures or to differences in the reference structures between subsequent models, due to actual anatomical changes or artifacts related to the low quality of the image at the area of interest [11]. Furthermore, for both methods, the selected reference areas contain non-osseous structures, which could also contribute to differences between the superimposed models. These factors are expected to have an increased impact on the outcomes when the superimposition reference area is relatively small [26].

A previous study [18] evaluating the intra- and inter-operator superimposition error of the TACB superimposition area found that the superimposition error increases if the distance between the measurement area and the superimposition reference area increases. This was also seen in all cases in this study, regardless of the superimposition reference area, as illustrated in Figure 1. However, the error for distant areas was still insignificant, except for a few cases in the MACB superimposition. It may be speculated that this pattern of error is related to the rotation of the T1 surface model, due to inadequate best-fit matching with the T0 model, during the superimposition process. Thus, the surface model rotates around an axis located somewhere within the reference area.

### Limitations

The assessment of surface model differences was performed using Viewbox 4 Software, the use of which has been repeatedly validated in previous studies [11,18,23,26,27]. Thus, no error is expected from this source.

The present sample consisted of CBCT volumes of growing patients only. It is well known that the midline cranial base area is morphologically stable from an early age [6,7,20,21,22], and thus, the MACB error cannot be attributed to this. Thus, our findings regarding both superimposition techniques are expected to be applicable in older patients as well.

It should be noted here that the reliability assessment performed in this study, namely, the visual inspection of the overlap of the stable anterior cranial base structures, following serial volume superimposition, has a solid scientific rationale, but it might not be considered highly precise.

Finally, the CBCT images included in the present study were acquired using the same machine and settings, which eliminates this confounding factor, but on the other hand it might limit the generalizability of the findings. The present images represent regular-quality CBCT images that are used for craniofacial morphology assessments. It remains to be tested if the present findings are modified by significant variations in image quality.

## 5. Conclusions

The anterior cranial base voxel-based superimposition methods applied here on a group of growing individuals, using two different reference areas (TACB and MACB), showed good overall reproducibility. The two areas showed good agreement to each other; however, the reliability of the MACB method was reduced in certain cases, and when individual case measurements were assessed, reduced reproducibility, as well as disagreement with the TACB method, were evident. For this reason, we recommend the extended anterior cranial base structures (TACB) for the 3D assessment of the craniofacial changes with the tested method.

## Figures and Tables

**Figure 1 jcm-10-05429-f001:**
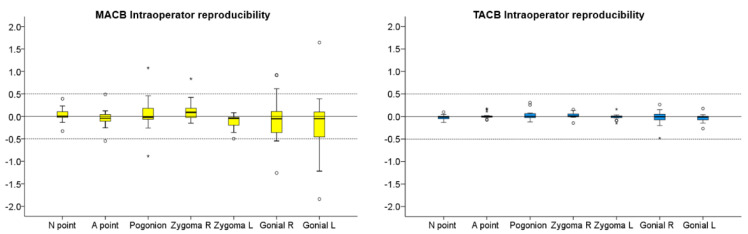
Box plots showing the intra-operator reproducibility of the MACB (Middle Anterior Cranial Base) and the TACB superimpositions (Total Anterior Cranial Base) for the detected T0–T1 changes in mm, for all measurement areas. The zero value, depicted by the continuous horizontal line, indicates perfect reproducibility, whereas any deviation from zero is considered an error. The dashed lines indicate 0.5 mm and −0.5 mm. In the boxes, the upper limit of the black line represents the maximum value, the lower limit the minimum value, the box the interquartile range, and the horizontal black line the median value. Outliers are shown as black dots or stars in more extreme cases.

**Figure 2 jcm-10-05429-f002:**
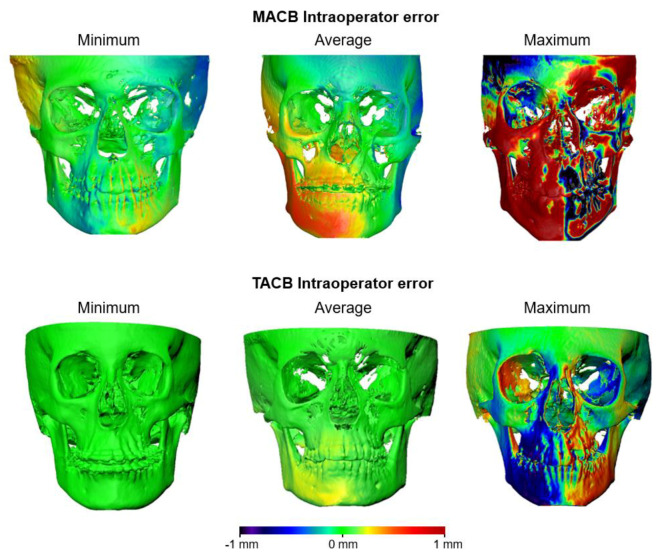
Color-coded distance maps showing the intra-operator differences on T1 surfaces obtained from repeated T0–T1 MACB and TACB voxel-based superimpositions, with the T0 surface held constant as a reference. The samples that presented the least (**left**), average (**middle**), and largest (**right**) absolute differences on the seven measurement areas are shown.

**Figure 3 jcm-10-05429-f003:**
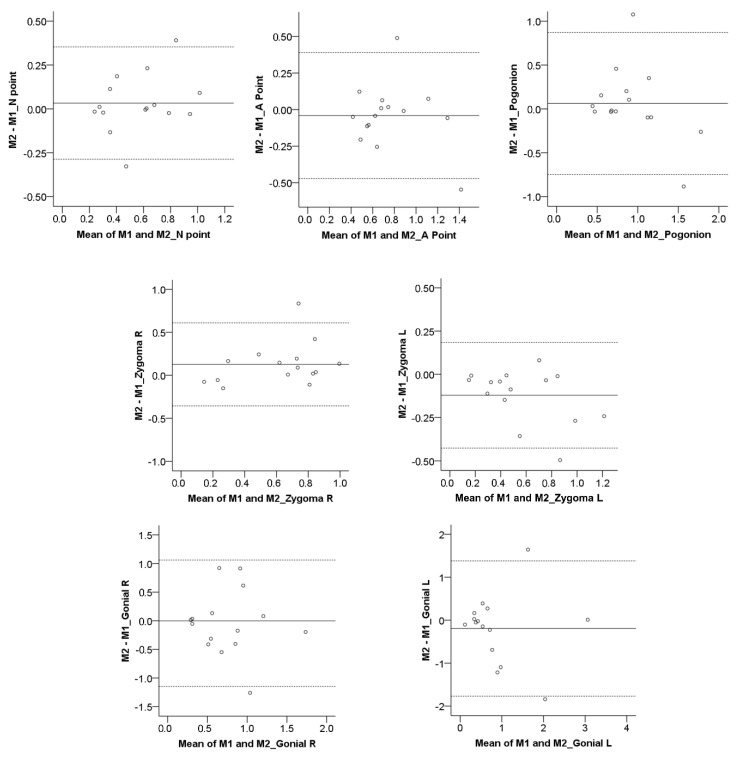
Bland–Altman plots of the T0–T1 changes (mm) detected through repeated MACB superimpositions by the same operator. The continuous horizontal line shows the mean of the differences in the detected T0–T1 changes, and the dashed lines show the corresponding 95% Limits of Agreement. M1: Measurement 1; M2: Measurement 2.

**Figure 4 jcm-10-05429-f004:**
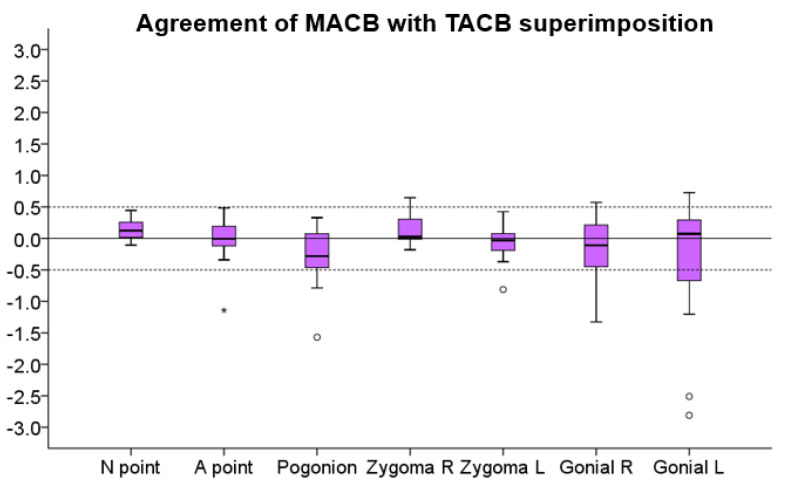
Box plots showing the agreement of the MACB (Middle Anterior Cranial Base) with the TACB (Total Anterior Cranial Base) superimposition for the detected T0–T1 changes in mm, for all measurement areas. The zero value, depicted by the continuous horizontal line, indicates perfect agreement, whereas any deviation from zero is considered a disagreement. The dashed lines indicate 0.5 mm and −0.5 mm. In the boxes, the upper limit of the black line represents the maximum value, the lower limit the minimum value, the box the interquartile range, and the horizontal black line the median value. Outliers are shown as black dots or stars in more extreme cases.

**Figure 5 jcm-10-05429-f005:**
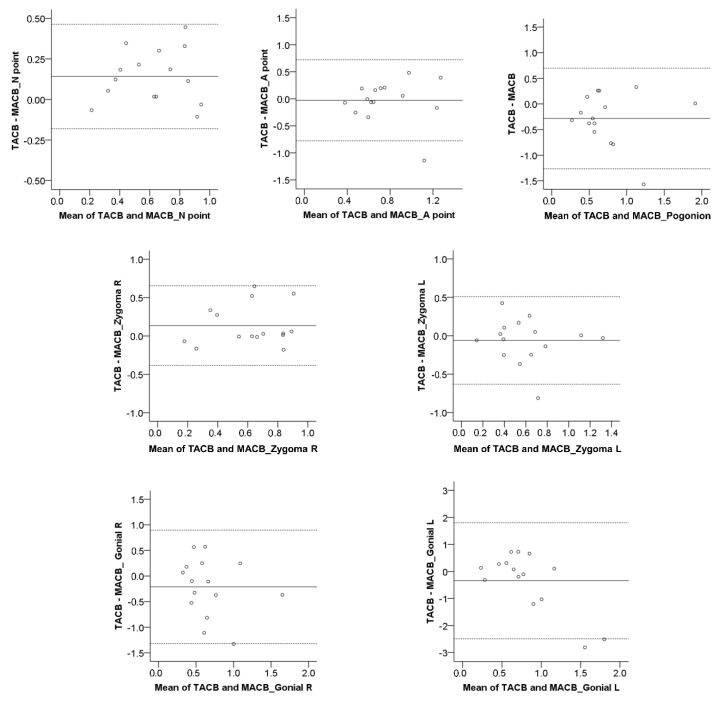
Bland–Altman plots of the T0–T1 changes (mm) detected through the TACB and the MACB superimpositions by the same operator. The continuous horizontal line shows the mean of the differences in the detected T0–T1 changes, and the dashed lines show the corresponding 95% Limits of Agreement. TACB: Total Anterior Cranial Base; MACB: Middle Anterior Cranial Base.

**Figure 6 jcm-10-05429-f006:**
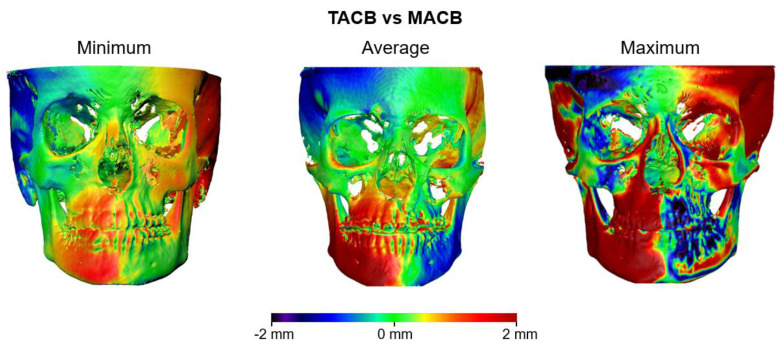
Color-coded distance maps showing the differences of T1 surfaces obtained from T0–T1 TACB and MACB voxel-based superimpositions, with the T0 surface held constant as a reference. The samples that presented the least (**left**), average (**middle**), and largest (**right**) absolute distances of the MACB from the TACB T1 surface, on the seven measurement areas, are shown.

## Data Availability

All data are available in the main text or the extended data. The protocols and datasets generated and/or analyzed during the current study are available from the corresponding author on reasonable request.

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
