# Peer review of "Reliability of Different Anterior Cranial Base Reference Areas for Voxel-Based Superimposition"

_jcm, 2021, doi:10.3390/jcm10225429_

Round 1

Reviewer 1 Report

1- The firts group whoo used voxel based superimposition was a group from unoiversity of Michigain. It was not first intorduced in a case report and there is alot of literature from Cevidanes et al is missing in the introduction.

2- Please mention the name of the software used by Weishmer at al

3- How come a prospective design? if the data is already existing?

4- The range of the height for the superimpostion box seems to be alot. 1 cm? 

5- why were not the T1 models exported as dicoms oriented on T0 and then compare the differences in both T1 models from both methods?

Reviewer 2 Report

1. It is regarded as a significant study that contributed to the development of an important approach for patient follow-up studies.

2. Materials and procedure
During CT scans of patients, a more detailed explanation is required to determine the relationship between the maxilla and the mandible. The mandible's reference point might differ considerably depending on the relationship setting.

Because references include many articles by the same co-author and corresponding author, it is advised that the number of references by the same author be reduced.

Reviewer 3 Report

Very well conducted study, with solid scientific methodology and very nicely presented results with representative graphics. I have no further suggestions to the authors.

This is a very useful and well conducted study. The authors describe a reliable and user friendly method for CBCT superimposition, which is becoming part of the standard of care for growth assessment in orthodontic and/or orthognathic surgery patients. The methodology and statistic analyses are solid and well described. The results are analyzed and presented in detail, with the additional use of beautiful graphics. In the discussion, the limitations of the study are fully acknowledged by the authors. However, they do not seam to compromise the value of the results of this study. Overall, this is  a high quality paper, with  impact in the clinical practice.
